# High-Sensitivity Temperature Sensor Based on Fiber Fabry-Pérot Interferometer with UV Glue-Filled Hollow Capillary Fiber

**DOI:** 10.3390/s23187687

**Published:** 2023-09-06

**Authors:** Yiwen Zheng, Yongzhang Chen, Qiufang Zhang, Qianhao Tang, Yixin Zhu, Yongqin Yu, Chenlin Du, Shuangchen Ruan

**Affiliations:** Key Laboratory of Advanced Optical Precision Manufacturing Technology of Guangdong Higher Education Institutes, Sino-German College of Intelligent Manufacturing, Shenzhen Technology University, Shenzhen 518118, China; 2110416042@stumail.sztu.edu.cn (Y.Z.); 2060413005@stumail.sztu.edu.cn (Y.C.); 13457787846@163.com (Q.Z.); 2210412014@stumail.sztu.edu.cn (Q.T.); 2210412023@stumail.sztu.edu.cn (Y.Z.); ruanshuangchen@sztu.edu.cn (S.R.)

**Keywords:** Fabry-Pérot interferometer, fiber temperature sensor, high-temperature sensitivity, ultraviolet glue cavity

## Abstract

Optical fiber Fabry-Pérot (FP) interferometer sensors have long been the focus of researchers in sensing applications because of their simple light path, low cost, compact size and convenient manufacturing methods. A miniature and highly sensitive optic fiber temperature sensor using an ultraviolet glue-filled FP cavity in a hollow capillary fiber is proposed. The sensor is fabricated by fusion splicing a single-mode fiber with a hollow capillary fiber, which is filled with ultraviolet glue to form a FP cavity. The sensor has a good linear response in the temperature testing and high-temperature sensitivity, which can be increased with the length of the FP cavity. The experimental results show that the temperature sensitivity reaches 1.174 nm/°C with a high linear response in the range of 30–60 °C. In addition, this sensor is insensitive to pressure and can be highly suitable for real-time water temperature monitoring for ocean research. The proposed ultraviolet glue-filled structure has the advantages of easy fabrication, high-temperature sensitivity, low cost and an arbitrary length of capillary, which has broad application prospects for marine survey technology, biological diagnostics and environmental monitoring.

## 1. Introduction

Temperature is a crucial physical parameter in the fields of the chemical industry, biological diagnostics and environmental monitoring. Furthermore, high-accuracy temperature sensing puts forward higher requirements for traditional temperature measurement. Compared with traditional temperature sensors, optical fiber temperature sensors have attracted great research interest due to their advantages in terms of their compact structure, flexible configuration, remote sensing capability, high sensitivity and immunity to electromagnetic interference [1]. A great number of temperature sensor structures have been created based on different sensing principles, including optical fiber grating [2,3], optical fiber interferometer and microstructured fiber, such as photonic crystal fiber [4,5,6], tapered fiber [7] and D-type fiber [8]. The sensitivity of fiber grating is relatively low and microstructured fiber involves an expensive and complicated configuration procedure. Therefore, interferometric temperature sensors have been widely used in research measurements within different fields.

The optical fiber interferometers in applying temperature sensing mainly include the Fabry-Pérot interferometer (FPI) [9,10], the Mach-Zehnder interferometer [11,12], the Michelson interferometer [13] and the Sagnac interferometer [14,15]. Among them, temperature sensors based on FPI are widely reported in practical applications owing to their simple principle, rapid response, convenient manufacturing methods and good mechanical properties. However, FPI sensors based on the silica material of fiber have low-temperature sensitivities of less than 20 pm/°C due to their low thermal expansion coefficient (TEC) of 5.5 × 10^−7^/°C and their low thermal optic coefficient (TOC) of 1.1 × 10^−5^/°C of silica material [16]. To enhance the sensitivity of FPI temperature sensors, the sensors can be fabricated by replacing the end surface in the FPI structure with polymer materials that have high thermo-sensitivity, such as polydimethylsiloxane (PDMS) [17,18], ultraviolet (UV) glue [19], polystyrene and polymethyl methacrylate [20]. Among the above materials, UV glue is a colorless, transparent and non-toxic liquid. Owing to their advantages of simple operation, fast curing and low cost, the various structures of temperature sensors with UV glue have been researched. Yinggang Liu et al. proposed a dual-parameter fiber-optic sensor structure, which consisted of an extrinsic FPI in the form of hemispherical UV curing glue capped on a fiber Bragg grating (FBG) end face and had a temperature sensitivity of 223.4 pm/°C in a range from 30 °C to 110 °C [21]. Jin Zhang et al. fabricated two cascaded FPIs for temperature sensing through the Vernier effect; the sensing FPI with an air cavity was composed of a cleaved fiber end-face and UV glue, while the reference FPI was fabricated by splicing single-mode fiber (SMF) with hollow capillary fiber (HCF) [22]. However, the system of grating inscription is expensive and complicated, and achieving the Vernier effect requires the fabrication of two sensor structures with matched cavity lengths by precisely controlling the reference FPI. Bowen Li et al. demonstrated a high-sensitivity temperature sensor based on a UV glue-filled silica capillary tube to reach 0.963 nm/°C [23]. Chengling Lee et al. presented a structure where the HCF was filled with a section of UV glue and an air gap remained between the SMF and UV glue; this sensor had a temperature sensitivity of −1.7 nm/°C [24]. Although both structures have high-temperature sensitivity, the UV glue cavity length is difficult to precisely control by dipping UV glue in HCF along the narrow gap under the action of capillary force. Meanwhile, in these aforementioned sensing schemes, they are either disturbed by pressure or do not mention the relevant measurement of pressure. In this paper, we propose a high-sensitivity fiber-optic temperature sensor based on the UV glue-filled Fabry-Pérot (FP) cavity in the HCF. The sensor is fabricated by fusion splicing a SMF with a HCF. Furthermore, this sensor is insensitive to pressure, which can be applied to temperature monitoring without interference from pressure; for example, temperature measurement in a water environment. The proposed sensor has the advantages of a compact structure, easy fabrication, good repeatability, high sensitivity and low cost.

## 2. Sensor Configuration and Principle

The proposed FP sensor structure is shown in Figure 1, which is manufactured by splicing a section of SMF and HCF. The HCF region is completely filled with UV glue, which is the sensing region of the proposed sensor. The refractive index (RI) of air, the SMF core and the UV glue are *n*_0_, *n_s_*, and *n*, respectively. Two reflected surfaces labeled as M_1_ and M_2_ are formed on a UV glue FPI with a cavity length of *L*. The RI of the cured UV glue is 1.48. The reflectivity of the two surfaces is *R*_1_, *R*_2_, which can be calculated by the Fresnel reflection equation [25] and can be expressed as:
(1)$$R_{1}={\left(\frac{n-n_{s}}{n+n_{s}}\right)}^{2},R_{2}={\left(\frac{n-n_{0}}{n+n_{0}}\right)}^{2}$$

Because of the low reflectivity of each surface, which is less than 4%, the higher-order reflections from these surfaces may be ignored [26]. The FPI can be regarded as double-beam interference to analyze the sensor structure. When the light emitted from the lead-in SMF passes through M_1_, part of the light is reflected, and part of the light is transmitted into the FP cavity, where the transmitted light will also be reflected and transmitted on the other reflective surface M_2_ of the FP cavity after a certain loss. Finally, the two beams of reflected light passing through M_1_ interfere due to the phase difference. Figure 2 illustrates the double-beam reflection model of the electric fields of the proposed sensor. The total electric field reflected from the FPI sensor $E_{r}$ can be expressed as [27,28]:(2)$$E_{r}=E_{0}\sqrt{R_{1}}+E_{0}\left(1-R_{1}\right)\left(1-\alpha \right)\sqrt{R_{2}}e^{-2j\Phi }$$
where $E_{0}$ is the input electric field, *α* is the transmission loss at the M_1_, *λ* is the wavelength of incident light and *Φ* is the phase difference in the FP cavity, which is defined as:(3)$$\Phi =\frac{4\pi nL}{\lambda }$$

The total reflected light intensity $I_{r}$ can be described by the square modulus of the intensity ratio of the reflected electric field to the incident electric field and can be given by:(4)$$I_{r}=|E_{r}/E_{0}{|}^{2}=R_{1}+R_{2}{\left(1-\alpha \right)}^{2}{\left(1-R_{1}\right)}^{2}+2\sqrt{R_{1}R_{2}}(1-\alpha )(1-R_{1})cos\left(\frac{4\pi nL}{\lambda }\right)$$

The wavelength of the *m*′th-order (*m* is a positive integer) interference dip of FPI can be obtained by
(5)$$\lambda_{m}=\frac{4nL}{2m+1}$$

The free spectral range (*FSR*) of FPI can be expressed as:(6)$$FSR=\frac{\lambda^{2}}{2nL}$$
the temperature sensitivity of the proposed FPI can be calculated by:(7)$$S_{T}=\frac{\Delta \lambda_{m}}{\Delta T}=\lambda_{m}(\frac{dL}{LdT}+\frac{dn}{ndT})$$

According to the formula, the temperature sensitivity of the sensor is determined by the TEC and the TOC. The TEC and TOC of the UV glue are 2.75 × 10^−4^/°C and 1.82 × 10^−4^/°C, respectively, which is two orders of magnitude for TEC and one order of magnitude for TOC larger than that of silica, so the TEC and TOC of the silica capillary can be negligible. Therefore, the temperature sensitivity can be simplified as:(8)$$S_{T}=\frac{\Delta \lambda_{m}}{\Delta T}=\lambda_{m}(\alpha_{TEC}+\sigma_{TOC})$$
where *α_TEC_* is the TEC of the UV glue, *σ_TOC_* is the TOC of the UV glue. Table 1 explains the meanings corresponding to all parameters in the paper.

## 3. Sensor Fabrication

The core and cladding diameters of SMF (SMF-28, Corning, New York, NY, USA) are 8.2 μm and 125 μm. The HCF has an inner diameter of 75 μm and an outer diameter of 125 μm. The key method for making the proposed sensor is UV glue filled in the HCF, which has an air core. To fabricate the proposed temperature sensor based on FPI with UV glue-filled HCF, a conventional fusion splicer (Fitel, Carrollton, GA, USA, S179) and a fiber cleaver (Sumitomo electric, Osaka, Japan, FC-6) are required. Figure 3 shows a schematic diagram of the manufacturing process of the sensor probe, which involves four steps. Step 1: A section of HCF is fused to the cleaved end-face of the lead-in SMF with a lower discharge power (25 bits) and shorter discharge duration time (500 ms), which can guarantee the fusion piece does not collapse and provides enough connection strength between the SMF and HCF to keep the splicing end face smooth, as shown in Figure 3a. The quality of the spliced joint can be evaluated by observing the insertion loss of spectrum after splicing and the process has nice reproducibility and dependability. Step 2: The HCF is cut to a designed length with the fiber cleaver. A Charge Coupled Device (CCD) camera is used for real-time monitoring of optical fiber movement to obtain the ideal length of HCF in general, as shown in Figure 3b. Step 3: A drop of UV glue is filled into the HCF. With the help of the capillary effect, the UV glue gradually flows into the interior of the HCF. We use the fusion splicer to observe the whole filling process. Another cleaved SMF is dipped into the UV glue. The end-face of the fiber with a small UV glue droplet is attached to the end of the HCF with a lateral offset. Due to the presence of air, UV glue slowly flows into the HCF and, after 5–10 min, is completely filled and there is no air in the HCF, as shown in Figure 3c. Finally, the sensor probe is completed after irradiating the UV glue using ultraviolet radiation for 20 min, as shown in Figure 3d.

The microscope image of the sensor probe is shown in Figure 4. To research the effect of the UV glue-filled length on the interference spectrum, we fabricated three samples (marked by S_1_–S_3_) with different cavity lengths for comparison. The lengths of the capillary were 69.92, 87.19 and 118.85 μm. Figure 5 presents the reflection spectra and microscope images corresponding to three samples. According to reflection spectra, the FSR of three different cavity lengths can be obtained to be 12.02, 9.35 and 7.18 nm, respectively. The FSR of the FPI decreased as the length increased, which was consistent with the results of (6). By comparing the FSR at different wavelengths, as shown in Figure 5, it can be seen that the experimentally obtained FSR increased with the increasing wavelength.

## 4. Results and Discussion

### 4.1. Temperature Performance

The temperature or gas pressure experimental setup of the proposed sensor is illustrated in Figure 6. In order to investigate the temperature sensing performance with different cavity lengths, we experimented with samples S_1_–S_3_ in a temperature response test. A broadband light source (BBS, Golight, Ultra-wideband Light Source, Culbertson, NE, USA) with a wavelength ranging from 1250 to 1650 nm was used as the input light source. The transmitted light from BBS was introduced to the FPI sensor via an optical circulator. The reflection spectrum of the sensor was detected using an optical spectrum analyzer (OSA, AQ6370D) with a resolution of 0.02 nm. The FPI sensor was placed in a high-precision temperature-controlled chamber with a minimum step of 0.1 °C. During the temperature test, the temperature-controlled chamber was set from 30 °C to 60 °C with an interval of 5 °C. The FPI was heated and cooled between 30 °C and 60 °C and two groups of data of reflection spectra were recorded using the OSA after a steady temperature of each set for 5 min. We performed a linear fitting between the dip wavelength and temperature for the heating and cooling process. However, the linear fittings in the heating and cooling process have a large deviation, as shown in Figure 7 for sample S_1_. The deviation was derived from the residual stress of the cured UV material, because the curing process was graded, the UV lamp vertically irradiated the sensor probe in fabrication, which caused UV glue to be unevenly cured. The interior of the cured UV material had enough residual stress to generate strain, which can affect the temperature sensing measurement. Therefore, the fabricated sensor structure was annealed in a tube furnace with a temperature maintained at 60 °C for 5 h to release the residual stress in the interior of the cured UV material, which filled into the HCF.

Then, we remeasured the reflection spectra of the three samples S_1_–S_3_ at each temperature after annealing. The reflection spectrum of sample S_3_ is shown in Figure 8a, where the temperature rises from 30 °C to 60 °C with an interval of 5 °C. Along with the increase in temperature, the interference fringe shifts to the longer wavelength direction. Because the values for the TOC and TEC of the UV glue are positive, it can be obtained that $\frac{dL}{dT}$ and $\frac{dn}{dT}$ are positive. Therefore, both the RI and the length of the FPI cavity increase when the temperature rises, which causes the wavelengths to have a redshift according to (5). The three samples S_1_–S_3_ were tested for heating and cooling process, and the shift in wavelength exhibits a linear relationship with the increase in temperature. The error bar of samples S_1_–S_3_ represents the deviation in wavelength during the heating and cooling processes, and the linear fitting between the dip wavelength and temperature is shown in Figure 8b–d. After linear fitting, the sensitivity to temperature for samples S_1_–S_3_ can be obtained to be 0.999, 1.033 and 1.174 nm/°C, respectively. From Figure 8, we can see that the temperature sensitivity of the proposed sensor increases with the cavity length of FPI.

In order to study the effect of the capillary inner diameter on the sensitivity of the sensor, we used three types of HCF with inner diameters of 75, 30 and 25 μm, respectively, the outer diameters of which were all 125 μm. The microscope images of the end-face of the HCF with different inner diameters are shown in Figure 9. To compare the experimental results with the 75 μm inner diameter, we fabricated two more fiber structures, marked S_4_ and S_5_. The inner diameters and lengths of the HCF for sample S_4_ are 30 μm and 69.99 μm, respectively, and for sample S_5_, they are 25 μm and 87.15 μm, respectively. We tested the temperature response in the same way as mentioned above. The insets in Figure 10 display the spectral response of the temperature variation in detail. The linear fitting between the dip wavelength and the temperature results of the two samples are shown in Figure 10. The sensitivity to temperature for samples S_4_ and S_5_ can be seen to be 0.517 and 0.562 nm/°C, respectively. In order to directly analyze the sensitivity of the proposed sensors with different inner diameters, we calculated the unit length sensitivity of samples S_2_, S_4_ and S_5_, the results of which were 0.01185, 0.00739 and 0.00645 nm/(°C·μm), respectively. The FPI sensor with a larger inner diameter had higher temperature sensitivity at the same length, owing to more UV glue being filled in the capillary, which induced a larger expansion in the temperature variation process.

### 4.2. Pressure Performance

The response of the proposed sensor to gas pressure was also experimentally explored. The sensor head was sealed in the gas chamber and the other end was connected to BBS and OSA through a circulator. The gas pressure experimental apparatus of the sensor is shown in Figure 6. The gas pressure was generated and measured by a high-precision pressure gauge (ConST-811) ranging from 100 kPa to 1 MPa with a step of 100 kPa. The section of the spectra response at room temperature is shown in Figure 11a, which shows that the reflective spectrum experiences almost no shift with increasing pressure. We chose two dip wavelengths (Dip 1 and Dip 2) and performed their linear fittings of wavelength with respect to the pressure, as shown in Figure 11b. The pressure sensitivity for Dip 1 and Dip 2 was 2.187 × 10^−5^, 1.271 × 10^−5^, respectively, and it can be considered that the proposed sensor is insensitive to pressure and suitable for real-time water temperature monitoring for ocean research.

Comparisons of the temperature sensing performances and characteristics between our UV glue-filled HCF sensor and other structured fiber sensors cited in the literature, including sensing sensitivity values, temperature ranges, costs, structure sizes as well as fabrication processes, are listed in Table 2. Among them, the sensitivity of our proposed sensor is obviously higher than most of those reported in the references. Compared to the other two sensors with higher sensitivity, our proposed sensor structure can precisely control the cavity length by cleaving desired lengths under the observation of the CCD. In addition, the proposed sensor also has the advantages of being small in size, easy to manufacture and repeatable production, making the sensor promising in terms of its application in the field of temperature measurement.

## 5. Conclusions

In summary, in this study, an optic fiber temperature sensor based on a UV glue-filled FP cavity in a HCF is proposed and demonstrated. The experimental results show that the sensor has a good linear response in a range of 30–60 °C and high-temperature sensitivity, which can be increased as the length of the FP cavity is enlarged. The sensitivity of the probe with an FP cavity length of 118.85 μm can reach up to 1.174 nm/°C. The FPI sensors can be simply fabricated with different lengths and inner diameters of the capillary, and have good repeatability. In addition, the sensor is insensitive to pressure and highly suitable for real-time water temperature monitoring in ocean research. With appropriate packaging, the proposed temperature sensor with high sensitivity and a compact structure has broad application prospects in marine survey technology, biological diagnostics and environmental monitoring. Due to the limitations of the experimental instruments, we only measured the temperature response of this sensor between 30–60 °C, and did not obtain the temperature sensing measurement in the low-temperature region. In future work, we will further test the proposed sensor at low or even sub-zero temperatures.

## Figures and Tables

**Figure 1 sensors-23-07687-f001:**
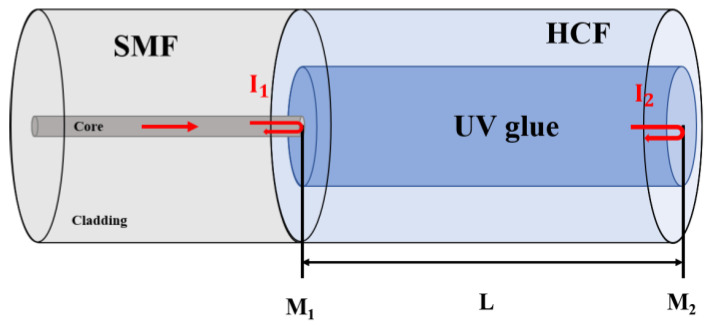
Schematic diagram of the proposed sensor structure.

**Figure 2 sensors-23-07687-f002:**
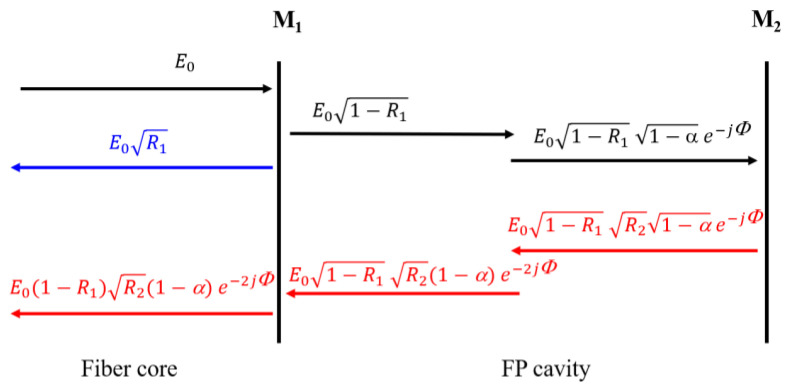
Two-beam interference model of the sensor.

**Figure 3 sensors-23-07687-f003:**
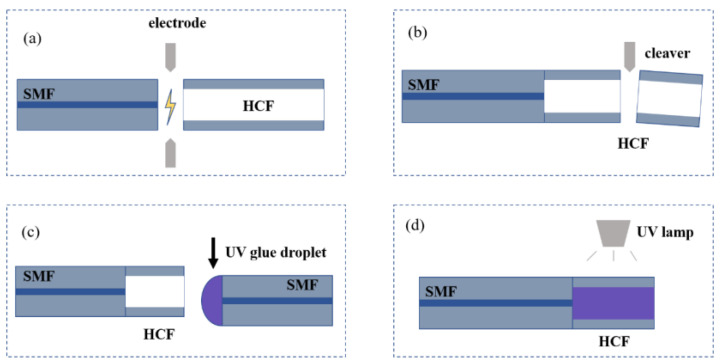
(**a**–**d**) Schematic diagram of the fabrication process of the FPI sensor probe.

**Figure 4 sensors-23-07687-f004:**
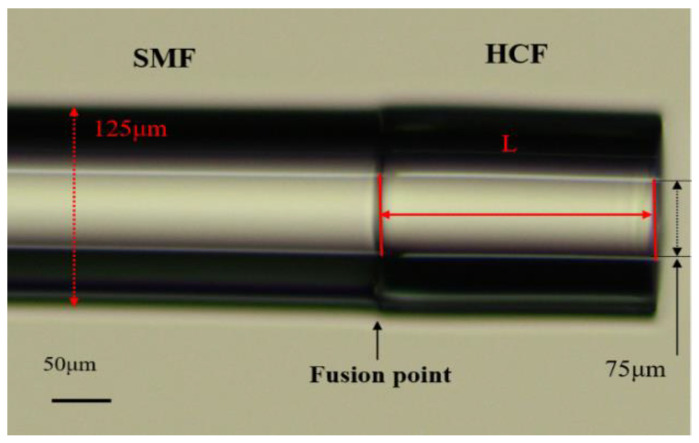
The microscope image of sensor probe.

**Figure 5 sensors-23-07687-f005:**
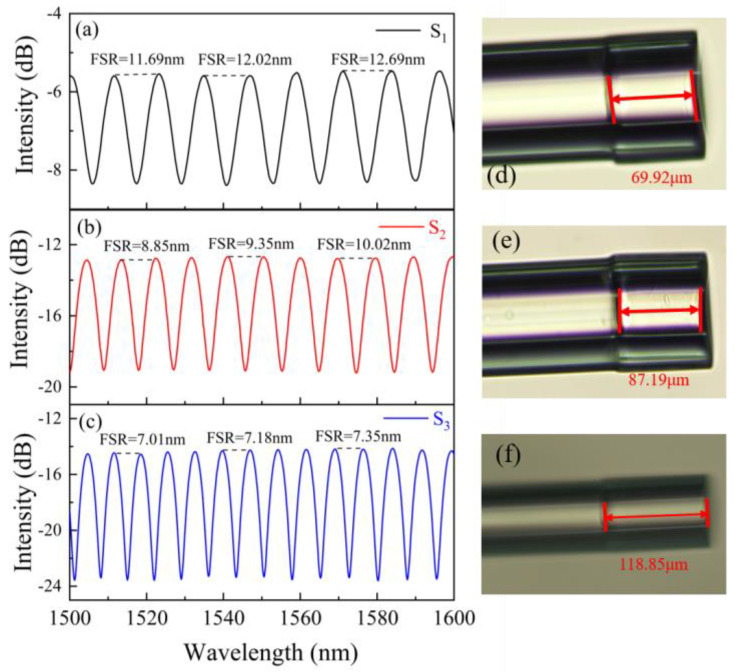
(**a**–**c**) Interference spectra of the proposed sensor S_1_–S_3_. (**d**–**f**) The microscope images of sensor structures with different cavity lengths corresponding to samples S_1_–S_3_.

**Figure 6 sensors-23-07687-f006:**
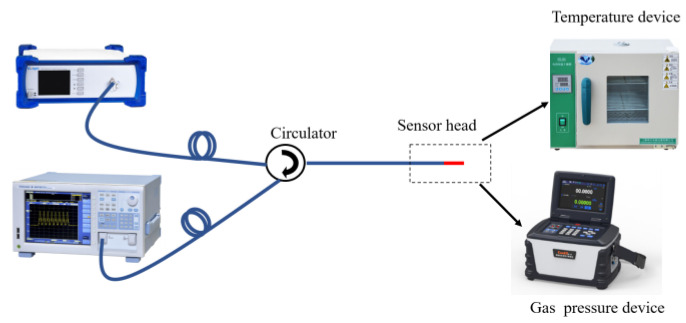
Experimental setup for temperature or gas pressure sensing measurement.

**Figure 7 sensors-23-07687-f007:**
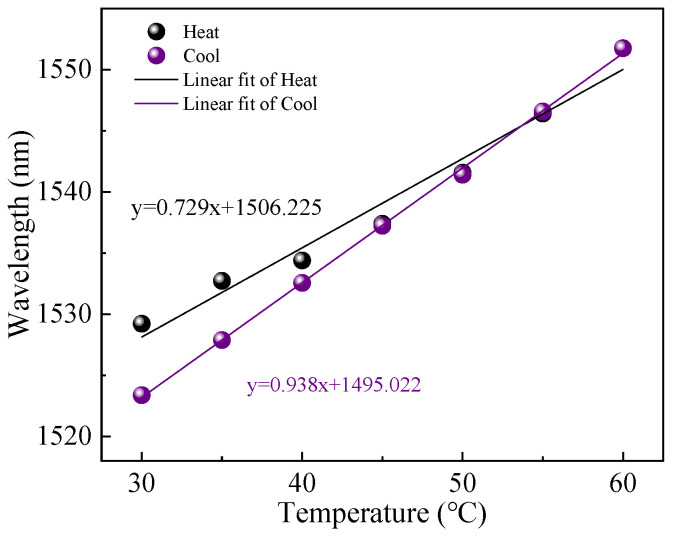
The relationship between the dip wavelength and temperature of sample S_1_ in the heating process and the cooling process.

**Figure 8 sensors-23-07687-f008:**
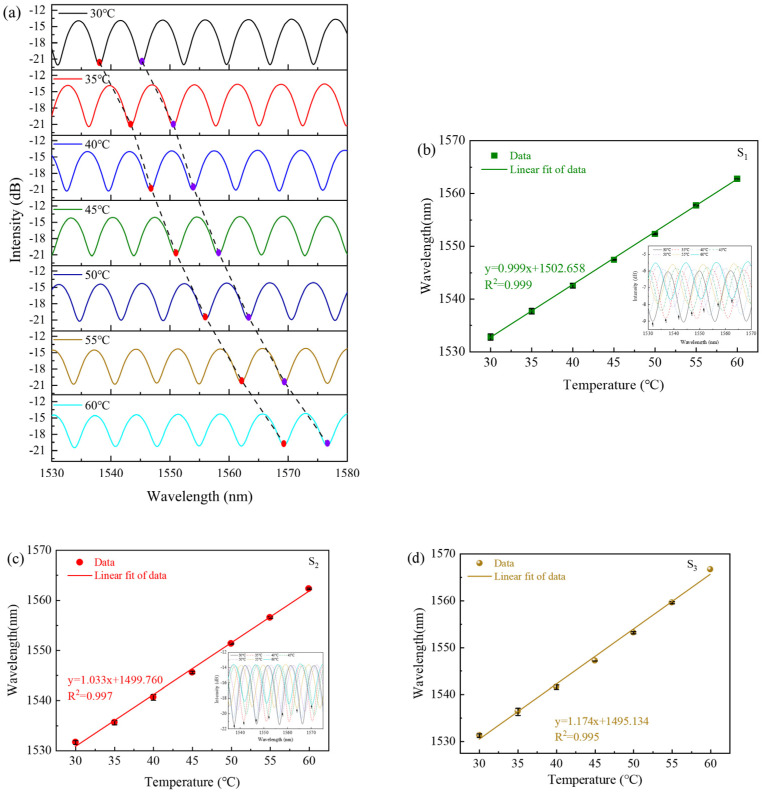
(**a**) The interference fringe of sample S_3_ with increasing temperature. (**b**–**d**) The linear fitting between the dip wavelength and temperature corresponding to samples S_1_–S_3_. Insets show their interference fringes with increasing temperature.

**Figure 9 sensors-23-07687-f009:**
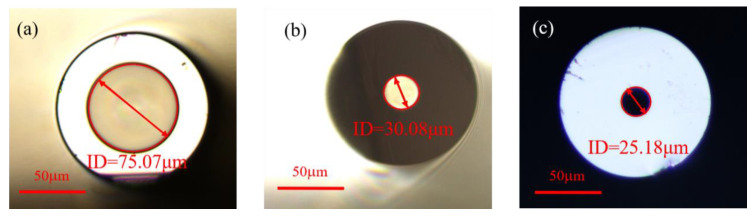
The microscope images of the end-face of the HCF with different inner diameters (ID): (**a**) 75 μm, (**b**) 30 μm and (**c**) 25 μm.

**Figure 10 sensors-23-07687-f010:**
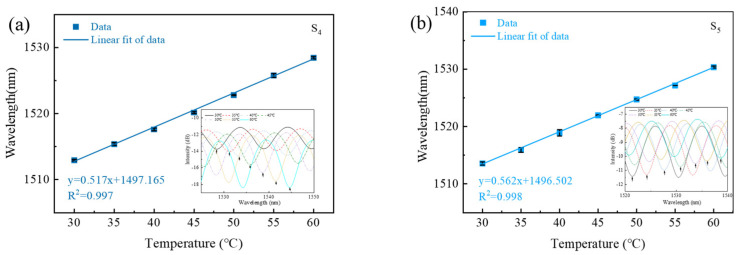
Linear fitting between the wavelengths and temperature variation with different inner diameters and lengths (**a**) S_4_ and (**b**) S_5_. Insets show their interference fringes with increasing temperature.

**Figure 11 sensors-23-07687-f011:**
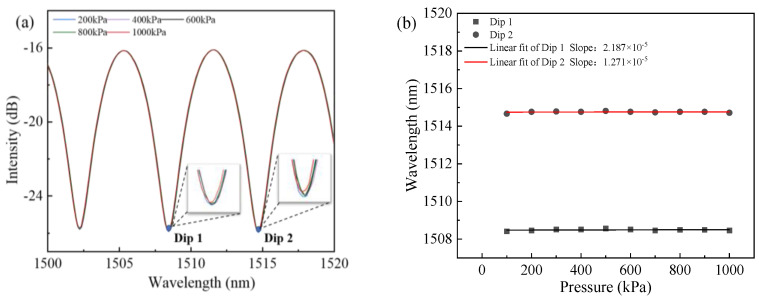
(**a**) The interference fringe with increasing pressure. (**b**) Wavelength shifts of Dip 1 and Dip 2 versus pressure.

**Table 1 sensors-23-07687-t001:** The parameters in the paper and the meaning of their explanations.

Parameter	Explanation
*n* _0_	RI of air
*n_s_*	RI of SMF core
*n*	RI of the UV glue
*R* _1_	reflectivity of reflected surface M_1_
*R* _2_	reflectivity of reflected surface M_2_
*L*	FPI cavity length
*α*	transmission loss at the M_1_
*λ*	wavelength of incident light
*Φ*	phase difference in FP cavity
*λ_m_*	wavelength of m′th-order interference dip of FPI
*E* _0_	input electric field
*E_r_*	total electric field reflected from the FPI sensor
*I_r_*	total reflected light intensity
*T*	temperature
*S_T_*	temperature sensitivity of the proposed FPI
*α_TEC_*	the TEC of UV glue
*σ_TOC_*	the TOC of UV glue
*m*, *j*	positive integer

**Table 2 sensors-23-07687-t002:** Sensing Performance Comparison of Several Different Temperature Sensors.

Sensor Structure	Sensitivity	FPI Cavity Length or Structure Size	Range	Fabrication	Cost	Ref
PDMS-filled air-microbubble	2.7035 nm/°C	the air-microbubble length of 26.5 μm	51.2–70.5 °C	Easy	Low	[17]
SU-8 and PDMS polymer-capped	0.6897 nm/°C	the SU-8 film thickness of 18.7 μm the PDMS film thickness of 7.7 μm	20–75 °C	Easy	High	[18]
UV glue-capped with FBG	0.223 nm/°C	the UV glue cap thickness of 48 μm	30–110 °C	Complicated	High	[21]
UV glue/Air cavity	67.35 nm/°C	the sensing air cavity length of 77.59 μm the reference air cavity length of 81.52 μm	20–24 °C	Complicated	Low	[22]
UV glue and Capillary	0.963 nm/°C	the UV glue cavity length of 90 μm	42–50 °C	Easy	Low	[23]
UV glue and Air Cavities	−1.7 nm/°C	the air cavity length of 20 μm the UV glue cavity length of 70 μm	20–75 °C	Easy	Low	[24]
UV glue-filled HCF	1.174 nm/°C	the UV glue cavity length of 118.85 μm	30–60 °C	Easy	Low	Our work

## Data Availability

No new data were created or analyzed in this study. Data sharing is not applicable to this article.

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
