# Peer review of "High-Sensitivity Temperature Sensor Based on Fiber Fabry-Pérot Interferometer with UV Glue-Filled Hollow Capillary Fiber"

_sensors, 2023, doi:10.3390/s23187687_

Round 1

Reviewer 1 Report

The authors, demonstrate a high sensitivity fiber-optic temperature sensor which was based on the UV glue-filled FP cavity in the HCF. The sensor is fabricated by fusion splicing a SMF with the HCF. In addition, this sensor is insensitive to pressure which can be applied to temperature monitoring without interference from pressure. The proposed sensor has the advantages of compact structure, easy fabrication, good repeatability, high sensitivity and low cost. I have a positive opinion about the manuscript and I find the manuscript is a good addition considering the scope of the Sensors journal. I recommend this manuscript for publication after addressing the following concerns within minor revision.

  • The title is clearly reflecting the content of the paper.
  • Abstract: the abstract effectively summarizes the manuscript and well understood.

·         Key words: were used properly.

  • In introduction part: The reasons for performing this study need to be clarified with more details.
  • Sensor Configuration and Principle parts: this parts are well presented. However, all parameters that are used in equations should be defined such as the parameter n in equation (2) and equation (5) need more explanation, as well as, all parameters that are used in experimental procedures should be mentioned in separate table. Furthermore, the authors mentioned “The TEC and TOC of UV glue are 2.75 × 10-4 /C and 1.82 × 10-4 /C 100, respectively”, add reference please. On the other hand, how the lengths of the capillary of (69.92, 87.19, and 118.85 µm) are selected or optimized, as well as, the authors stated “Therefore, both RI and length of FPI cavity increase when the temperature rises” needs to be double check.
  • The conclusion was clearly presented, however, the future work was not suggested.
  • The references, were used well and most of references are up to date.

Reviewer 2 Report

The presented work describes and characterizes a FP-based temperature sensors where the cavity has been filled with UV cured glue. Different sensors have been fabricated, with different cavity length and difference cavity diameter. The good sensitivity to temperature of the sensors have been demonstrated. Moreover, the authors have shown that the sensors are poorly influenced by pressure variation. 

Overall, the work is valid. The ideas are well described, the method and the experimental setup are well presented, and the results demonstrated the initial the claim on the authors. I find the work quite satisfactory, and I agree with publication.  

Reviewer 3 Report

The manuscript proposed for review is devoted to the development of a sensing element for temperature control based on the Fabry-Perot interferometer. The manuscript in general is of high quality, written in good scientific language. The authors suggest the idea of using UV glue as the sensing layer of the interferometer, and the HCF as its protective sheath. In spite of the fact that Fabry-Perot temperature sensors are not new, the manuscript as a whole is interesting. Despite the overall positive impression, I would like to make a few remarks. 

1- The abbreviations MZI, MI, SI were used only once in lines 43-44; PMDS and PMMA in lines 52-53; SCT in line 66. Further on in the text, they were not used. I recommend that the authors look carefully at the text and use abbreviations only for frequently used abbreviations. 

2. Line 98. The authors use TEC and TOC of the UV glue for sensitivity assessment. However, HCF and UV glue are combined into the single element that experiences a common temperature effect. The combination of HCF and UV glue cannot be substitute  using the properties of the UV glue. In addition, the HCF plus UV glue have different coefficients of thermal elasticity. Wouldn't the expansion of the UV glue cause damage to the HCF? This is not considered in any way in the manuscript. In this regard, the linear dependencies shown in Figures 7 and 9 are slightly questionable. I would ask the authors to provide a detailed explanation for this. 

3. Figure 2: It is best to sign “UV glue” instead of “UV” because UV is defined as “ultraviolet”, which is not a material. 

4 The term “FP cavity” is first used in line 71, but the abbreviation “FP” is not defined.

5. Equations (1)-(6). The authors have proposed the simplest model of the Fabry-Perot interferometer. This model is incomplete. I would recommend using a more complete mathematical model given, for example, in: https://doi.org/10.33640/2405-609X.3243.

6.  The interference spectra shown in Figure 4 are more similar to the spectra obtained from calculations using the formulas of the mathematical model than to the experimentally obtained spectra. The experimentally obtained FSR should increase with increasing wavelength, albeit slightly. This has not been investigated in the paper.

7. Figure 6. Please specify whether the deviation and fitting are related to the heating or cooling processes.

8. Line 166. Explain why an annealing process at 60 °C for 5 hours was chosen to release residual stress.

9. Line 213. What kind of gas was used to generate pressure and how it was introduced into the gas chamber?

10. Line 222. What is the average ocean temperature?

Reviewer 4 Report

This manuscript shows a low-cost easily-fabricated FP temperature sensor. Overall, it it well organized and technically completed. It could be better if the authors consider the followings: 

1. Put Table 1 in Introduction part. Use the same unit nm or pm per degree C for all listed sensitivities.

2. In equation (6), St has nothing to do with UV glue length. Why the temperature sensitivity of the proposed sensor was found to increase with the cavity length?

3. Any reason on choosing 69.92, 87.10, and 118.85 microns as cavity lengths? How you controlled the length or how you measured the length?

4. It seems only one heating and cooling cycle was tested? Did you try more than one cycles on one fiber to test its repeatability?

Round 2

Reviewer 3 Report

let it be